# Fast Perfekt: Regression-based refinement of fast simulation

**Moritz Wolf**[1*]**, Lars O. Stietz**[1,2†]**, Patrick L.S. Connor**[1,3]**, Peter Schleper**[1] **and Samuel Bein**[1,4‡]

**1** University of Hamburg, Institute of Experimental Physics, Hamburg, Germany
**2** Hamburg University of Technology, Institute of Mathematics,
Chair Computational Mathematics, Hamburg, Germany
**3** Center for Data and Computing in natural Sciences, Hamburg, Germany
**4** Université catholique de Louvain, Louvain-la-Neuve, Belgium

* moritz.wolf@uni-hamburg.de , † lars.stietz@tuhh.de , ‡ samuel.bein@uclouvain.be

## Abstract

The availability of precise and accurate simulation is a limiting factor for interpreting and forecasting data in many fields of science and engineering. Often, one or more distinct simulation software applications are developed, each with a relative advantage in accuracy or speed. The quality of insights extracted from the data stands to increase if the accuracy of faster, more economical simulation could be improved to parity or near parity with more resource-intensive but accurate simulation. We present Fast Perfekt, a machine-learned regression to refine the output of fast simulations that employs residual neural networks. A deterministic morphing model is trained using a unique schedule that makes use of the ensemble loss function MMD, with the option of an additional pair-based loss function such as the MSE. We explore this methodology in the context of an abstract analytical model and in terms of a realistic particle physics application featuring jet properties in hadron collisions at the CERN Large Hadron Collider. The refinement makes maximum use of existing domain knowledge, and introduces minimal computational overhead to production.

# 1  Introduction

The use of simulated data is essential in science and engineering to interpret and predict real-world data. Often, two simulation applications are developed to fulfill different purposes: one resource-intensive application (fullsim) that emulates real data as accurately as possible, and a faster application (fastsim) that produces large data sets with reduced computing overhead while sacrificing a degree of accuracy. The former (latter) is advantageous when bias (statistical precision) is a limiting factor. In fields such as climate and weather modeling, fullsim might predict regional ambient patterns whereas fastsim models global forecasts [1]. In precision measurements and searches for new physics with particle colliders, fullsim may be used for emulating background events [2], while 10 [3–5] or 100 [6] times faster fastsim is used for events corresponding to signal models with many free parameters [7, 8].

Significant computing and scientific benefits could be realized if fastsim were made highly accurate while maintaining its speed advantage. Recent approaches have been developed in particle physics to render data emulated by fastsim more accurate. For example, the DCTR [9] approach weights simulated data points such that their distributions more closely agree with target (fullsim) data, correcting distributions of features and their correlations. Limitations can arise, however, because the weights are specific to the underlying process, the support is limited to the domain of the input, and the weights reduce the statistical precision of the fastsim data. Other methods employ a Wasserstein metric or integral loss function to update simulated features via a mapping [10–12], the first using generative methods and the last two being deterministic mappings. These approaches can be effective in refining simulation without introducing weights. However, these methods typically do not exploit all relevant information present in the training samples, in particular the object-to-object correlations between fastsim and fullsim. Moreover, while probability density functions (PDFs) of the refined feature(s) may agree well with the target, the transformation may not be unique and can lead to degraded correlations with features not directly used by the network. More generally, ML has been used to replace components of fast simulation like the modeling of calorimeter showers, such as the seminal work [13], with a broad review given in [14]; similar techniques have been applied in ATLAS fastsim [3]. There are also efforts to replace the entire simulation frameworks with generative normalizing flows in CMS [15].

In this article, we propose Fast Perfekt, a deterministic regression-based approach to render the output features of fastsim uniquely consistent with fullsim. The relatively simple method employs two similar training data sets, a fastsim input set and a target fullsim set, to train a refiner network that morphs fastsim features directly. The training procedure employs one or more loss functions, the primary maximum mean discrepancy (MMD) [16] loss, which measures the similarity between two PDFs, and optionally a secondary MSE loss to measure

sample-to-sample similarity. Fast Perfekt aims to provide refinement of fastsim that is

- accurate in both the bulk and tails of distributions;

- effective in modeling correlations among several available and hidden features;

- weightless, preserving statistical precision; and

- fast and deterministic to ensure efficiency and traceability.

Fast Perfekt makes maximum use of domain knowledge in two senses. First, it takes fastsim as a baseline and applies only a residual correction, avoiding the task of learning the target data from scratch. Second, Fast Perfekt makes use of hidden information embedded in the sample-based matching between the input and target data.

Initial concepts of this method were previously reported as an application to CMS fast simulation (FastSim) [17]. The concept and procedure for training the network are described in Section 2. The methods are demonstrated in Section 3 using an example based on an abstract analytical data set. Fast Perfekt is then applied in a realistic physics application in Section 4 featuring jet substructure variables in LHC experiments. Section 5 summarizes the main findings, impact and limitations.

## 2 Method

Two alternate training schema are proposed, depending on the nature of the training data: a single-stage training procedure using only an ensemble loss, and a 2-stage procedure making use of ensemble and pair-based losses. Two data sets are required to train the network: an input set processed with the fastsim and a target set processed with the fullsim. Often, the fastsim and fullsim share one or more identical algorithmic aspects, for example, the ground truth (GT). It is suggested to synchronize any random variables where possible, for example, by using the same GT, and preparing the data with a unique matching between each fastsim and fullsim sample. When such synchronization is possible, the 2-stage training is suggested; otherwise, the single-stage schema has to be employed.

Let $\mathbf{x}'$ be a set of fastsim features to be refined. These features, or correlations among them, presumably exhibit inaccuracies with respect to the fullsim features $\mathbf{x}$. Each element of $\mathbf{x}'$ is taken as input to the refiner network along with any conditioning variables, such as the GT values $\mathbf{g}$ of the given object, together defining the vector $\mathbf{a}' = (\mathbf{g}, \mathbf{x}')^{\mathsf{T}}$. Additional features $\mathbf{h}'$ are hidden, either unavailable or not chosen among the set to be refined, but may also be important for downstream or upstream analysis. The network outputs refined features $\hat{\mathbf{x}}$ whose properties and correlations to the hidden features $\mathrm{corr}(\hat{\mathbf{x}}, \mathbf{h}')$ are estimators of the fullsim features $\mathbf{x}$ and correlations $\mathbf{corr}(\mathbf{x}, \mathbf{h})$, respectively. We note that the network goal is to render fastsim features more fullsim-like, and not more like the GT values; we also note that the network cannot refine the hidden variables themselves, but only correlations to them. A simple schematic of the sampling and training procedure is given in Figure 1.

### 2.1 Network architecture

The network architecture is inspired by the ResNet model [18], as shown in Figure 2. The input to each *residual block* is added back to its output via skip connections, which reduces the job of the network to determining residual corrections to the fastsim features. This is suitable for refinement problems where the fastsim already provides a reasonable approximation of the fullsim, including its stochastic dimensionality. Each skip block consists of two hidden layers for model flexibility, and we employ fully-connected (FC) linear layers in the residual blocks.

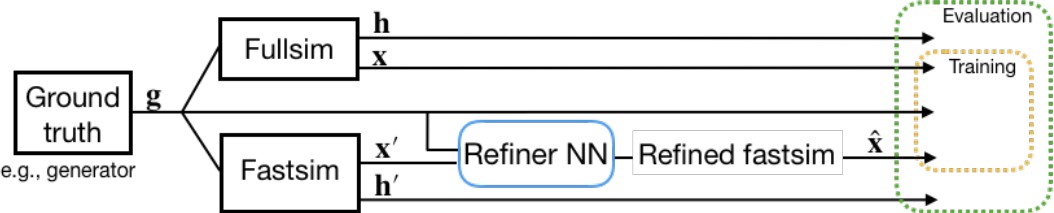

Figure 1: A schema of the Fast Perfekt training and evaluation. The fullsim features $\mathbf{x}$ and $\mathbf{h}$ and fastsim features $\mathbf{x}'$ and $\mathbf{h}'$ share the same ground truth $\mathbf{g}$. The refiner network is trained to apply a residual correction to obtain the refined fastsim data set $\hat{\mathbf{x}}$. The hidden features are used for an evaluation meta-study (green box) but are not incorporated into the training procedure (yellow box).

The weights and biases are initialized according to the Fixup method [19] such that before training, the network behaves as the identity function and returns the fastsim features: The last FC layer of each residual block and the final layer of the network are initialized to have zero weights and biases; the first layers of the network and of each residual block are initialized using the Kaiming-initialization [20]. The number of skip blocks $n_B$ and nodes per internal layer $n_L$ are adjusted to ensure sufficient network flexibility for accurate refinement. The values of the network hyperparameters used in the two studies are shown in Table 3, and other details specific to the individual study are discussed in Sections 3 and 4.

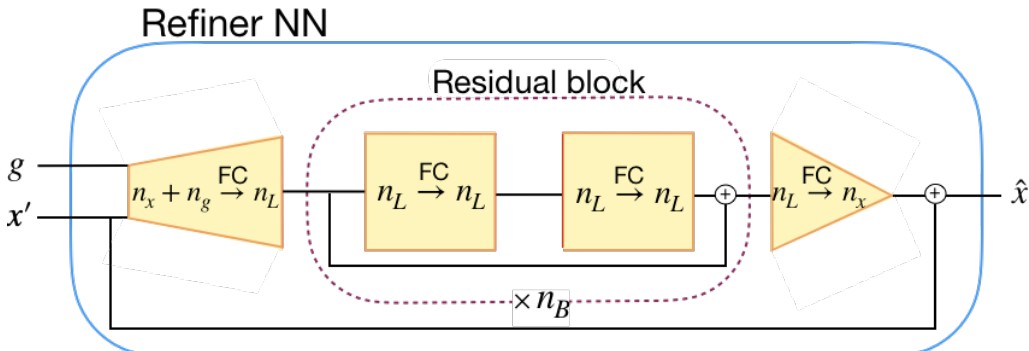

Figure 2: Architecture of the refiner neural network (NN). The yellow elements represent fully connected layers (FC). A residual block is made of two FC layers and a skip connection. The $n_{\{x,g,L\}}$ numbers correspond to the dimensions of the corresponding vectors and $n_B$ is the number of residual blocks.

## 2.2 Loss functions

The loss functions measure the similarity between the target data set $A = \{\mathbf{a_i}\}_{i=1,...m}$ and output data set $\hat{A} = \{\hat{\mathbf{a_i}}\}_{i=1,...m}$, where $\mathbf{a}_i = (a_i^1, ..., a_i^{n_a})$ and $\hat{\mathbf{a}}_i = (\hat{a}_i^1, ..., \hat{a}_i^{n_a})$, with data set size $m$ and number of dimensions $n_a$. The primary loss, which compares the multidimensional PDFs of the fullsim and refined fastsim data, is the MMD [16]. We employ the biased estimator

$$\text{MMD}_{\text{b}}(\theta) = \frac{1}{m^2} \sum_{i,j=1}^{m} k(\mathbf{a_i}, \mathbf{a_j}) + \frac{1}{m^2} \sum_{i,j=1}^{m} k(\hat{\mathbf{a_i}}(\theta), \hat{\mathbf{a_j}}(\theta)) - \frac{2}{m^2} \sum_{i,j=1}^{m} k(\mathbf{a_i}, \hat{\mathbf{a_j}}(\theta)) \tag{1}$$

with Gaussian kernel function

$$k(\mathbf{a}, \hat{\mathbf{a}}) = \exp\left(-\sum_{l=1}^{n_a} \frac{(\hat{a}^l - a^l)^2}{\sigma_l^2}\right), \tag{2}$$

where $\theta$ is the vector of trainable network parameters and $l$ is the dimension index. The bandwidths of the kernel function (2) are set to the median Euclidean distance between the target and input in the respective dimension, $\sigma_l = \text{median}\{\|a_i'^l - a_j^l\|_2 : i, j \in [m]\}$ [16]. The single-stage recipe and the second stage of the 2-stage recipe use only this loss. The 2-stage method also makes use of the unbiased estimate $\text{MMD}_u$ given by

$$\begin{aligned}\text{MMD}_u(\theta) = &\frac{1}{m(m-1)} \sum_{i=1}^{m} \sum_{i \neq j}^{m} k(\mathbf{a_i}, \mathbf{a_j}) \\ &+ \frac{1}{m(m-1)} \sum_{i=1}^{m} \sum_{i \neq j}^{m} k(\hat{\mathbf{a}}_i(\theta), \hat{\mathbf{a}}_j(\theta)) - \frac{2}{m^2} \sum_{i,j=1}^{m} k(\mathbf{a_i}, \hat{\mathbf{a}}_j(\theta)),\end{aligned} \tag{3}$$

as well as the mean squared error (MSE)

$$\text{MSE}(\theta) = \frac{1}{m} \sum_{i=1}^{m} \|\hat{\mathbf{a}}_i(\theta) - \mathbf{a_i}\|_2^2. \tag{4}$$

The biased and unbiased MMD are highly correlated, but $\text{MMD}_u$ has properties that better facilitate convergence in the first training stage, due to its having a well-defined expectation value of 0 for two i.i.d. data sets. This condition is reached via a constrained optimization, where we employ the modified differential method of multipliers (MDMM) [21], which identifies saddle points in the space of the network parameters and Lagrange multiplier $\lambda$. Then, $\text{MMD}_b$ is better suited for refining the finest details because of its well-defined lower boundary at 0 corresponding to two ensembles in perfect agreement. Minimizing the MSE alone would result in significant biases due to regression to the mean. However, because it encodes information about correlations among visible and hidden features, minimizing the MSE conditionally while constraining $\text{MMD}_u$ to 0 serves to protect or restore correlations.

To assess the network performance more robustly, we incorporate information about the hidden features in the loss by including $\mathbf{h}$ and $\mathbf{h}'$ in the vectors $\mathbf{a}$ and $\hat{\mathbf{a}}$. This yields the so-called *omniscient* MMD, which is used for evaluation but not for training. We also examine binned ratios of the refined and target PDFs and a $\chi^2$ statistic derived from the comparison of these PDFs. These measures are not used in the training at any stage but only for studying the performance of the network.

## 2.3 Training

When the GT is not synchronized for the training and target samples, such as when simulation is being refined to better match real data, we suggest a single-stage training using $\text{MMD}_b$ as the only loss. When the samples are synchronized, we suggest a 2-stage prescription. In the first stage, the $\text{MMD}_u$ and MSE losses are used simultaneously to guide the network to a configuration close to its global optimum, and in the second stage $\text{MMD}_b$ is used on its own to bring the network into the minimum. Details of the 2-stage recipe are given in the following.

In the first stage, the MSE is minimized while constraining $\text{MMD}_u$ to 0, which is the expectation value for two i.i.d. distributions. The purpose of the first stage is to achieve maximum similarity between the associated sample pairs (MSE) while improving and preserving consistency between the output and target distributions ($\text{MMD}_u$).

The Lagrangian used in the MDMM is

$$\mathcal{L}(\theta, \lambda) = \text{MSE}(\theta) - \lambda\big(\varepsilon - \text{MMD}_\text{u}(\theta)\big) - \frac{\delta}{2}\big(\varepsilon - \text{MMD}_\text{u}(\theta)\big)^2 \qquad (5)$$

with $\varepsilon = 0$. The Lagrangian (5) is minimized with respect to the network parameters $\theta$ and maximized with respect to the Lagrange multiplier $\lambda$. The MSE and $\text{MMD}_\text{u}$ may be correlated at the start of training, but they eventually become anti-correlated as the network converges to a point along the Pareto front. The MDMM algorithm constrains the $\text{MMD}_\text{u}$ loss to a desired value along the front. The quadratic term in Equation (5) provides a damping effect with a constant parameter $\delta$.

After the MDMM converges, in the second stage of the training, only the $\text{MMD}_\text{b}$ loss is minimized with respect to the trainable network parameters, and the refiner is allowed to converge to its unconditional optimum. This second stage is needed to fine-tune the agreement between the fullsim and refined fastsim, particularly in the tails of distributions.

## 3 Analytical example

We consider an analytical two-dimensional data set based on simple expressions that play the role of GT, fullsim, and fastsim. The first dimension is the feature $\mathbf{x}$ to be refined. The second dimension is the hidden feature $\mathbf{h}$ and is not taken as input to the network nor to the training loss, but is included in the study to demonstrate the impact of the training method.

### 3.1 Data set

A single GT data set is synthesized by sampling a bimodal 2D Gaussian mixture probability density comprising the sum of two 2D Gaussian distributions with different means and covariance matrices, giving a large population $L$ and a small population $s$:

$$G_L \sim \mathcal{N}(\mu_L, \Sigma_L), \quad G_s \sim \mathcal{N}(\mu_s, \Sigma_s). \qquad (6)$$

Taking $m = 250000$ to be the total number of data samples $\mathbf{g} = (x^\text{GT}, h^\text{GT})^T$ used, a large population in the GT distribution is sampled from $G_L$ with $0.85m$ samples $\mathbf{g}_0, \ldots, \mathbf{g}_{0.85m-1}$ and a small population is sampled from $G_s$ with $0.15m$ samples $\mathbf{g}_{0.85m}, \ldots, \mathbf{g}_m$, without loss of generality.

To produce the fullsim and fastsim data sets, the GT data are smeared and shifted. Smearing is added based on two independent random variables with covariance $\Sigma_\text{fast}, \Sigma_\text{full}$ as

$$S_\text{fast} \sim \mathcal{N}(0, \Sigma_\text{fast}), \quad S_\text{full} \sim \mathcal{N}(0, \Sigma_\text{full}). \qquad (7)$$

For maximum generality, different bias values $b^\text{fast}$ and $b^\text{full}$ are added *to the small populations* for the fastsim and fullsim. A summary of the values defining the analytical data set are documented in Table 4 in the Appendix. The analytical data set is shown in Figure 3.

### 3.2 Training

A ResNet-like model as described in Section 2.1 is trained with hyperparameters given in Table 3 in the Appendix. We present the results after training with only the MSE loss, with the single-stage, and with the 2-stage prescriptions described in Section 2.3.

The results of training using only the MSE loss are shown in Figure 4 (left column). As anticipated, regression to the mean effects are observed. The modified fastsim underestimates the tails and overestimates the bulk of the distributions. Importantly, however, this network

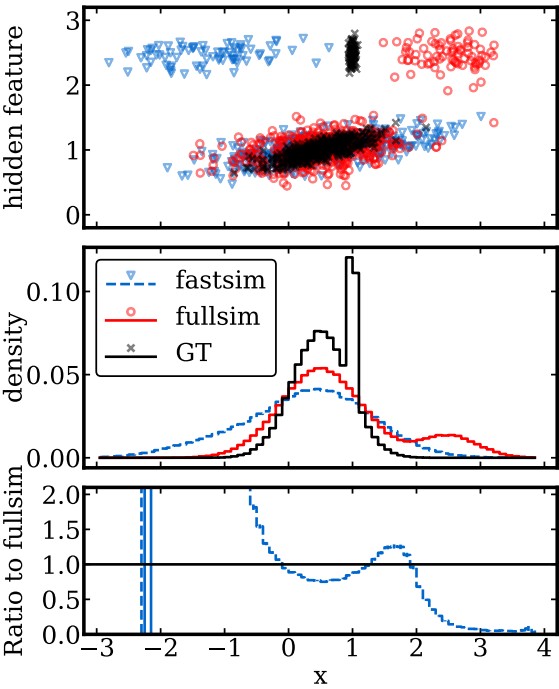

Figure 3: Distributions of the data sets including the unrefined fastsim, target fullsim, and GT. The sub-figures show a scatter plot of the complete information (top), a histogram of the feature to be considered for refinement (center), and a histogram of the bin-by-bin ratio of fastsim counts to fullsim counts (bottom). For the scatter plot a random subset of 500 data points is considered for illustration purposes.

correctly assigns the centers of the two populations to their target, thus removing the bias in original fastsim related to the hidden feature.

The results of training with only the $\text{MMD}_b$ loss are shown in Figure 4 (middle column). The agreement in the projection of the refined feature appears to be improved. However, the populations are not correctly assigned to their correct positions based on the target data. The network uses data points from the large population to compensate for the small population and fills the hole resulting in the large population with data points from the small population. In other words, the correlations between the available and hidden features remain broken even when the training has converged to the minimum $\text{MMD}_b$ value.

Each loss brings its own advantage and disadvantage. Minimizing MSE loss alone results in accurate bias correction but poor modeling of the target PDF, whereas minimizing MMD alone results in a good modeling of the refined PDF but poorly models the correlation.

Figure 4 (right column) shows the results of the 2-stage training outlined in Section 2. First, the MSE and $\text{MMD}_u$ are minimized simultaneously using the MDMM algorithm, which learns to fulfill the constraint $\text{MMD}_u = 0$. After around 30 epochs, the network has converged, and the second stage begins, where the $\text{MMD}_b$ alone is minimized. This network results in well-modeled bulk and tails exceeding the accuracy of the network trained only with MMD, as can be seen in Table 1, and also results in the two refined populations being correctly matched to the target fullsim. The omniscient MMD, which measures the accuracy while accounting for the refined as well as hidden features, is significantly lower after the 2-stage training procedure than after either of the single-loss versions.

In conclusion, after minimizing only the MMD, good refinement can be achieved when examining the space of the refined feature. However, the converged network is not unique, and is sensitive to the initialization of the weights. By incorporating a pair-based loss (MSE) in

the initial training stage via the MDMM algorithm, the final network is constrained to render equivalent samples more similar, which leads to more accurate correlations between the refined and hidden feature. The network learns correlations between the available and hidden features through the MSE because the matching criterion implicitly carries information about the unseen features. The benefits of the MSE are maximal when the biased features occupy relatively empty regions of the domain.

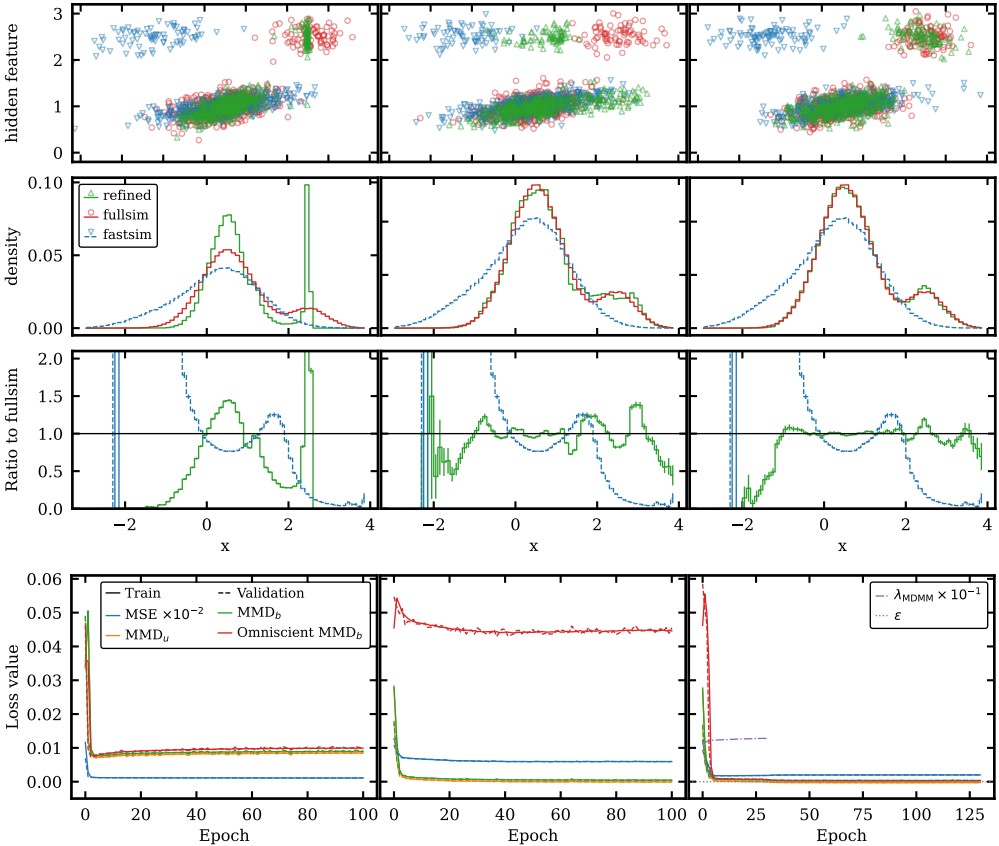

Figure 4: Distributions of the fastsim, refined fastsim, and fullsim data sets, as well as loss curves for the MSE-only training (left), the MMD-only training (middle), and the 2-stage training (right). For the scatter plots a random subset of 500 data points is considered for illustration purposes.

# 4   Application to collider physics

The refinement protocol using the 2-stage training scheme introduced above is applied to a realistic example tailored to high-energy physics data collected at the LHC. The data set is described, choices regarding the training process are discussed, and the results of the refinement application follow.

## 4.1   Data set

Analysis of LHC data relies on the accurate simulation of both particle collisions and the reconstruction of produced particles. In proton-proton collisions at the LHC, particles of various types are produced, and the final states often feature one or more jets, which are sprays of col-

Table 1: Results of the refinement applied to the analytical example. The degree of refinement is quantified by the lowering of the (omniscient) MMD, MSE and $\chi^2$/ndof compared to the original fastsim.

| Fullsim vs. | Fastsim | Refined (MMD-only) | Refined (2-stage) |
|---:|:---:|:---:|:---:|
| Omniscient $\text{MMD}_b \times 10^3$ | $38.39 \pm 2.90$ | $45.11 \pm 3.03$ | $0.33 \pm 0.17$ |
| $\text{MMD}_b \times 10^3$ | $39.07 \pm 3.70$ | $0.41 \pm 0.30$ | $0.23 \pm 0.15$ |
| $\text{MSE} \times 10^3$ | $1384 \pm 63$ | $602 \pm 24$ | $200 \pm 6$ |
| $\chi^2$/ndof | $3493$ | $28$ | $20$ |

limated hadrons. Correctly simulating the production and reconstruction of jets is of high importance across a wide range of investigation, for example, unfolding measurements [22, 23] as well as searches for physics beyond the standard model, e.g. Refs. [24, 25]. Likewise, searches for new heavy states often probe for processes with highly boosted decay products that merge into so-called fat jets [26], which exhibit distinct or exotic substructure properties.

Jets are notoriously difficult to model because their production mechanisms are only partly calculable with perturbative methods and because the energetic response of the detectors is a complicated function of the detector geometry, event activity, and radiation exposure. Considerable effort has been made to accurately model jets in state-of-the-art fullsim models such as GEANT4-based [27–29] applications of ATLAS and CMS. Fullsim processing time per event ranges from 10s of second to a few minute per event, depending on the underlying physics process. In fastsim programs, such as ATLAS AtlFast3 [3], CMS FastSim [4, 5], the processing time is only a few seconds, but due to simplified assumptions in the calorimetry and tracking detectors, mismodeling of jets is often compounded. In parametric simulators, such as the Delphes framework [6], the processing time per event ranges in the milliseconds, but jet mismodeling arises due to simplified energy response parametrizations.

In this section, we examine the potential for Fast Perfekt to refine mismodeled fastsim jets. This method does not generate or modify intermediate representations of jets or their energy deposits in the detector but functions directly on analysis-level observables. Incorporating Fast Perfekt increases the CPU time per event by around one millisecond, negligible compared to current fastsim and around half of the processing time of Delphes.

We consider a set of jet substructure observables called N-subjettiness $\tau_N$ [30]. Ratios such as $\tau_2/\tau_1$ provide information on the compositeness of the jets to infer the unobservable particles from which they originated. A GT data set is generated based on proton-proton collisions at $\sqrt{s} = 13$ TeV using the PYTHIA 8.1 software package [31], consisting of events with a pair of top quarks in the final state. These events are then processed twice in parallel using Delphes, once with the default CMS detector implementation and treated as the fullsim data set, and once with a "flawed" implementation yielding the data set we treat as the fastsim. The flaw is introduced by setting the $\beta$ parameter [30] to 0.9 rather than 1.0, inducing an angle-dependent bias in the N-subjettiness, and setting the energy resolution of the electromagnetic calorimeter to a constant of 1% rather than the default values, which depend on transverse momentum $p_T$ and pseudorapidity $\eta$ [32, 33].

The data set used for training consists of approximately 1 million GT jets and their respective fastsim and fullsim jets, which are associated to each other based on their trajectories. It is split into train, validation, and test data sets consisting of 160, 40, and 40 batches of 4096 jet triplets, each constituting a matched GT, fullsim, and fastsim jet.

## 4.2 Training

The refinement network is trained as described in Section 2 using the hyperparameters given in Table 3. The refinement is applied to the three-dimensional space of N-subjettiness ratios whose fastsim versions are given to the network as inputs along with their respective GT values:

$$\mathbf{x}' = \left( \frac{\tau_2}{\tau_1}, \quad \frac{\tau_3}{\tau_2}, \quad \frac{\tau_4}{\tau_3} \right)^{\mathsf{T}} \tag{8}$$

$$\mathbf{g} = \left( \frac{\tau_2^{GT}}{\tau_1^{GT}}, \quad \frac{\tau_3^{GT}}{\tau_2^{GT}}, \quad \frac{\tau_4^{GT}}{\tau_3^{GT}} \right)^{\mathsf{T}}. \tag{9}$$

Additionally, 6 hidden observables are part of the data set and are used for evaluation: the jet mass, $p_{\mathrm{T}}$, $\eta$, the distance to the closest neighbor jet $\mathrm{dR} = \sqrt{\mathrm{d}\eta^2 + \mathrm{d}\varphi^2}$ (with azimuthal angle $\varphi$), and the numbers of charged and neutral jet constituents $N(\mathrm{ch})$ and $N(\mathrm{ne})$. For the calculation of the omniscient MMD, a $\log_{10}$ transformation is applied to the jet mass and $p_{\mathrm{T}}$.

Figure 5 shows the evolution of the loss functions, MSE, $\mathrm{MMD_b}$, and $\mathrm{MMD_u}$, together with the Lagrange multiplier $\lambda$ and the omniscient $\mathrm{MMD_b}$ as evaluation metric. For the first stage, a global factor of 10 is applied to $\mathrm{MMD_u}$ to make its order of magnitude similar to that of the MSE loss. The switch to stage two happens when the MDMM algorithm has converged, which is the case after 118 epochs.

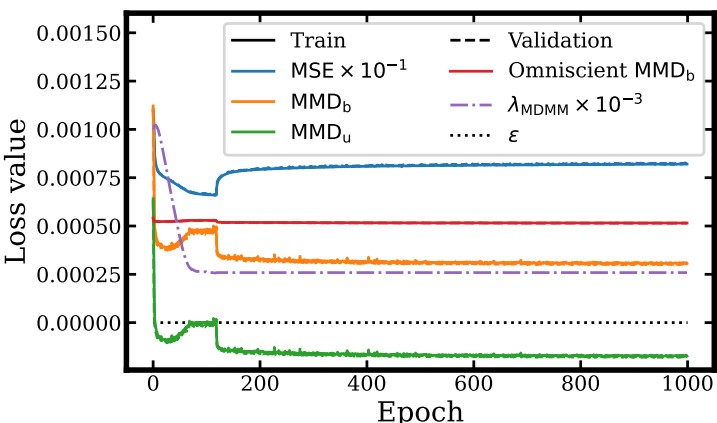

Figure 5: The MSE, (omniscient) MMD, and $\lambda$ during the training. The switch to stage two (MMD-only) after epoch 118 is visible as kinks in the curves.

## 4.3 Results

Figure 6 shows the distributions of the N-subjettiness ratios for the three sets of jets: fullsim, original fastsim, and refined fastsim. It is apparent that the refinement leads to a more accurate simulation both in the bulk and tails of the distributions. Evaluating the performance beyond one-dimensional projections, the top row in Figure 7 shows the Pearson correlation coefficients within a set of variables consisting of both the variables seen by the network and the hidden observables for fullsim (left), fastsim (center), and refined fastsim (right). The bottom row shows for each cell the absolute difference to the respective fullsim value. The refinement again leads to a consistent improvement in the fastsim modeling. Notably, correlations between the visible and hidden variables improve, although they are not known to the network. Quantifying the accuracy in terms of several metrics, Table 2 shows the MMD and omniscient MMD before and after refinement, as well as the MSE and a $\chi^2$ measure. The $\chi^2/\mathrm{ndof}$ is derived from a binned ratio of the N-subjettiness in fullsim and refined fastsim;

statistically independent validation data sets are used for this purpose and the mean $\chi^2$ is taken from among the three dimensions. The relatively small change in omniscient MMD can be explained by the fact that most of the dimensions ($g$ and hidden) remain unchanged by the refinement.

Table 2: Results of the refinement applied to the collider physics example. The degree of refinement is quantified by the lowering of the (omniscient) MMD, MSE and $\chi^2$/ndof compared to the original fastsim. The MMD values and errors are calculated as the respective means and standard deviations of all batches from the validation data set.

| Fullsim vs. | Fastsim | Refined (MMD-only) | Refined (2-stage) |
|---:|:---:|:---:|:---:|
| Omniscient MMD$_b \times 10^3$ | $0.5386 \pm 0.0077$ | $0.5147 \pm 0.0068$ | $0.5149 \pm 0.0066$ |
| MMD$_b \times 10^3$ | $1.114 \pm 0.087$ | $0.305 \pm 0.022$ | $0.303 \pm 0.024$ |
| MSE $\times 10^3$ | $10.99 \pm 0.23$ | $8.25 \pm 0.02$ | $8.24 \pm 0.02$ |
| $\chi^2$/ndof | $33.34$ | $1.97$ | $2.52$ |

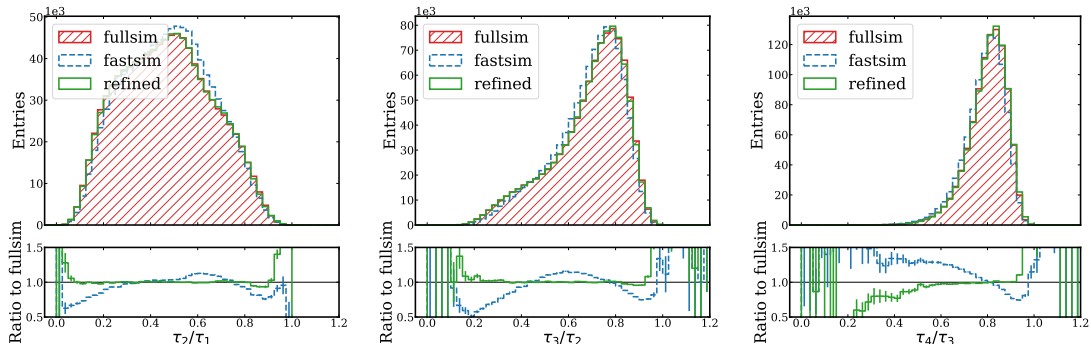

Figure 6: Distributions of the N-subjettiness ratios for fullsim, original fastsim, and refined fastsim. The improvement resulting from the refinement becomes apparent in the ratio to fullsim (lower panels).

Also shown in Table 2 are the accuracy measures based on a single-stage MMD training. The performance of the two prescriptions is similar, first of all suggesting overall robustness of the method. Additionally, the single-stage prescription is sufficient for the variables considered in this application, and in general when the modes of the unrefined FastSim and target distributions already coincide well. A single-stage training where simulation is refined to match real data may therefore be highly optimal, particularly if the simulation is somewhat performant. If applicable, however, the 2-stage training is more robust in that the scope of hidden features is a priori unknown, and because it leads to a more constrained network.

## 5   Conclusions

We have introduced a regression-based procedure for refining the final analysis variables produced by a fast simulation application (fastsim) to render them highly accurate with respect to cost-intensive full simulation (fullsim). The refinement model, which is based on a residual neural network, is trained by minimizing an ensemble loss (MMD), which evaluates the similarity between two unbinned multi-dimensional distributions. As an additional option, a secondary pair-based loss (MSE) is used in a 2-stage training prescription: one stage that

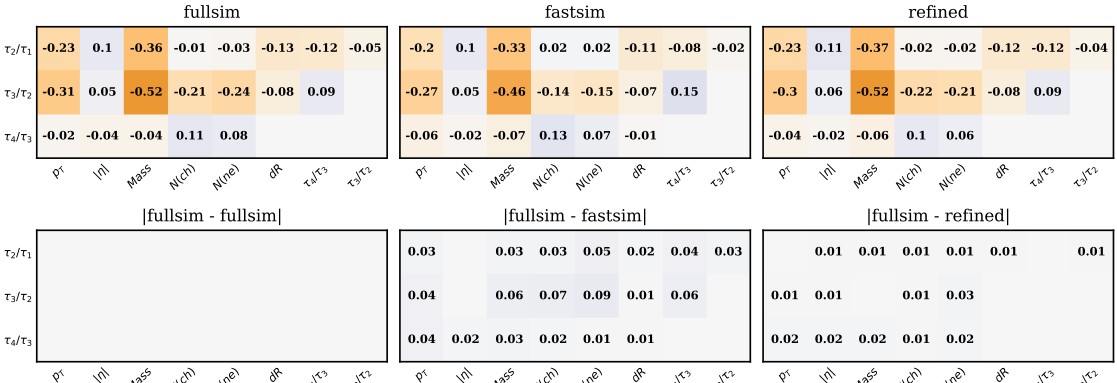

Figure 7: Pearson correlation coefficients rounded to two digits (top row) within the set of visible and hidden variables for fullsim (left), original fastsim (center), and refined fastsim (right). The bottom row shows the absolute differences of these correlation coefficients to fullsim.

combines the MMD and MSE losses, and a second stage which uses only the MMD.

Two examples have been explored for refinement, each considering the accuracy of features that are available to the network as well as hidden features which are not. An abstract example refines fastsim and fullsim proxy data sets defined analytically and synchronized at the level of the ground truth. We find that the single-stage MMD-only procedure performs well when evaluating the accuracy of the available feature; however, it falls short in describing the multidimensional target space which includes the hidden dimension. Introducing a 2-stage prescription that makes use of the MSE loss improves the accuracy, improving the description in the multidimensional domain. A second example based on a realistic model of collider physics at the CERN LHC examines the performance of the two prescriptions with a larger number of available and hidden features. Here, the 2-stage prescription leads to similar quality of refinement as the single-stage prescription.

Fast Perfekt makes comprehensive use of existing domain knowledge present in fastsim programs, minimizing the need for the network to learn most of the salient features and correlations present in the target data. The refinement acts on final variables or summary statistics of the simulation (i.e., jet properties) rather than intermediate representations (i.e., calorimeter shower hits). The utility of the MMD as a loss function for regression has been demonstrated, and the MMD is deemed suitable for refinement problems that target real-world rather than simulated data. In cases where the input and target samples can be synchronized at the level of the ground truth, the 2-stage Fast Perfekt prescription is preferred, as it constrains the network to a highly unique configuration by making use of the MSE loss. This increases the stability and overall accuracy of the refined simulation, and is considered ideal for tuning fastsim to match fullsim.

## Acknowledgements

**Funding information**   M.W. is funded by the Deutsche Forschungsgemeinschaft (DFG, German Research Foundation) under Germany's Excellence Strategy – EXC 2121 „Quantum Universe" – 390833306.  L.S. acknowledges financial support from grant HIDSS-0002 DASHH (Data Science in Hamburg - Helmholtz Graduate School for the Structure of Matter). P.L.S.C.'s work was supported by University of Hamburg, HamburgX grant LFF-HHX-03 to the Center

for Data and Computing in Natural Sciences (CDCS) from the Hamburg Ministry of Science, Research, Equalities and Districts, and by the BMBF under contracts U4606BMB1901 and U4606BMB2101.

# A  Appendix

Table 3: Network hyperparameters and training setup for the trainings on the analytical data set and the application to collider physics.

|  | Analytical data | Collider physics application |
|---|---|---|
| Input dimensions | 2 | 6 |
| Output dimensions | 1 | 3 |
| Residual blocks | 2 | 5 |
| Nodes per layer | 64 | 512 |
| Activation function | LeakyReLU ($\alpha = 0.01$) | |
| Optimizer | ADAM | |
| Loss function | MSE / MMD / MDMM | |
| MMD kernel bandwidth(s) | 1.3658 | (0.0351, 0.0183, 0.0058, 0.0415, 0.0207, 0.0043) |
| Learning rate network | $1 \times 10^{-4}$ | $5 \times 10^{-6}$ |
| Learning rate $\lambda_{\mathrm{MDMM}}$ | $1 \times 10^{-3}$ | $1 \times 10^{-4}$ |
| Initial $|\lambda_{\mathrm{MDMM}}|$ | 1 | 1 |
| Epochs | 100 (per stage) | 1000 (total) |
| Early stopping | 10(3) Epochs (MDMM) | No |
| Batch size | 2048 | 4096 |
| Data Set size | 250000 | 983040 |

Table 4: Parameters defining the analytical data set and their corresponding values. The bias terms $b$ are added only to the small populations.

| Parameter | Value |
|---|---|
| Total Number of Samples | $N = 250000$ |
| $\mu_L$ | $\begin{pmatrix} 0.5 \\ 1 \end{pmatrix}$ |
| $\Sigma_L$ | $\begin{pmatrix} 0.2 & 0.05 \\ 0.05 & 0.02 \end{pmatrix}$ |
| $\mu_s$ | $\begin{pmatrix} 1 \\ 2.5 \end{pmatrix}$ |
| $\Sigma_s$ | $\begin{pmatrix} 0.001 & 0 \\ 0 & 0.02 \end{pmatrix}$ |
| $\Sigma_{\text{fast}}$ | $\begin{pmatrix} 0.5 & 0.04 \\ 0.04 & 0.01 \end{pmatrix}$ |
| $\Sigma_{\text{full}}$ | $\begin{pmatrix} 0.02 & 0 \\ 0 & 0.02 \end{pmatrix}$ |
| $b_{\text{fast}}$ | $\begin{pmatrix} -2.2 \\ 0 \end{pmatrix}$ |
| $b_{\text{full}}$ | $\begin{pmatrix} 1.5 \\ 0 \end{pmatrix}$ |

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
