# Peer review of "Fast Perfekt: Regression-based refinement of fast simulation"

_SciPost Physics Core, doi:SciPost Phys. Core 8, 021 (2025)_

## Round 1 · Referee Report · Anonymous (Referee 1) · 2024-12-16

Strengths
- timely: precise and fast simulation is crucial for the success of the current and future generation of collider experiments
- light-weight: it provides a much more lightweight alternative to generative AI-based fast simulation,
- versatile: can be used on top of such generative models or "traditional" fastsimulation
Weaknesses
- examples seem very small (less than 5 dimensional), and it's unclear if the shown performance scales to other applications (where the refinement would be more important)
Report
The authors propose a machine-learning-based algorithm to morph samples from "fastsimulation" closer to samples from "fullsimulation". This provides a light-weight alternative to generative AI applications to fast simulation. Depending on whether or not a "pairing" of events from full- and fastsim based on common ground truth information is possible, the authors propose and investigate the use of two different loss functions and training strategies. The model shows good performance in the considered examples and therefore seems promising.
I think the manuscript should be published in SciPost, however, I have a few questions that I would like to see addressed first (see below).
Requested changes
- Is there a way to find out which loss to use in a non-toy case? I guess one always wants to include as many variables as possible, so there will not be an omniscient MMD available. I'm asking because I was wondering about the application to calorimeter simulation, where EM showers for the same incident conditions (incident energy / angle) are very similar, but hadronic showers (for example from pions) differ very much from shower to shower.
- When comparing $corr(x,h)$ to $corr(\hat{x},h')$, the results will be skewed if $h$ and $h'$ differ from each other. I miss a discussion on this in section 2 and also the toy case does not consider $h\neq h'$.
- Will the method still work if the Fastsim model suffers from mode-collapse? I'm asking because in the limit of arbitrarily close Fastsim-samples, a deterministic model will not be able to spread the samples apart.
- The use of the terms "fastsim" and "fullsim" in the LHC example is misleading, as both are essentially fastsim, just with different parameter cards.
Minor comments - The abbreviation MDMM is first used below equation 4, but only explained a few paragraphs below that, just above equation 5. - The paragraph below equation 6 uses once $N$, and $m$ otherwise. - The introduction of section 4 refers to "data collected at the LHC", when in fact only simulation is used in the following. - GEANT4 asks for 3 references (see https://geant4.web.cern.ch/ bottom)
Recommendation
Ask for minor revision
Dear reviewer,
Thank you for the thorough and helpful review, and for considering the paper to make a valuable contribution. Also, I am sorry I accidentally posted this response to the wrong set of comments.
We agree with the basis of all of the comments, and propose changes to the paper draft in most cases. We respond and propose below in line. As for the noted point regarding very high-dimensional cases such as the case of refining individual hits within a shower, we have realized the need to clarify the purpose of Fast Perfekt, at least in its presented form, as intended to act on final variables of the simulation and not on intermediate quantities such as hits; we clarify this below in responding to all items.
== Requested changes Reviwer: Is there a way to find out which loss to use in a non-toy case? I guess one always wants to include as many variables as possible, so there will not be an omniscient MMD available. I'm asking because I was wondering about the application to calorimeter simulation, where EM showers for the same incident conditions (incident energy / angle) are very similar, but hadronic showers (for example from pions) differ very much from shower to shower.
Authors: We propose to tweak the final paragraph to make it more clear that we endorse using the 1-stage, MMD-based training in cases where simulation is refined to match real data, and the 2-stage method is ideal to make fastsim more like fullsim. For intermediate objects like sim hits from EM showering, the loss terms would likely need to be restructured. Fast Perfekt’s purpose is really to morph the final analysis variables (summary statistics) of objects and events. We propose to add the following sentence in the last paragraph of the conclusions: “ The refinement acts on final variables or summary statistics of the simulation (i.e., jet properties) rather than intermediate quanities (i.e., calorimeter shower hits).”
Reviwer: When comparing corr(x,h) to corr(x^,h′), the results will be skewed if h and h′ differ from each other. I miss a discussion on this in section 2 and also the toy case does not consider h≠h′.
Authors: We agree, that we can improve the correlation to the hidden observable, but not to the extent that would be possible if one could also refine h. This case is explored in the Delphes example where the pT and other variables are included in the omniscient MMD but not directly refined. We propose to add a qualifying statement about correlations to hidden observables: “we also note that the network cannot refine the hidden variable itself, but only correlations to the target hidden variable.”
Reviwer: Will the method still work if the Fastsim model suffers from mode-collapse? I'm asking because in the limit of arbitrarily close Fastsim-samples, a deterministic model will not be able to spread the samples apart.
Authors: In an extreme case of mode collapse, where all elements of the feature vector default to a given value, Fast Perfket would not be able to refine such an effect away, because the refinement is deterministic and does not add its own noise. However, if some variables (or the ground truth) have not exhibited mode collapse, there is still the possibly to refine.
Reviwer: The use of the terms "fastsim" and "fullsim" in the LHC example is misleading, as both are essentially fastsim, just with different parameter cards.
Authors: We propose to modify the sentences on 243 for clarity to read: “These events are then processed twice in parallel using Delphes, once with the default CMS detector implementation and treated as the fullsim data set, and once with a
flawed" implementation yielding the data set we treat as the fastsim.”
System Message: WARNING/2 (<string>, line 27); backlink
Inline literal start-string without end-string.
Minor comments Reviwer: The abbreviation MDMM is first used below equation 4, but only explained a few paragraphs below that, just above equation 5.
Authors: We agree with the minor comments propose to properly define and cite the MDMM when it is first introduced. Reviwer: The paragraph below equation 6 uses once N, and m This typo has been corrected
Reviwer: The introduction of section 4 refers to "data collected at the LHC", when in fact only simulation is used in the following.
Authors: We propose to change “based on” to “tailored to” so it can not be interpreted that we have used real data.
Reviwer: GEANT4 asks for 3 references (see https://geant4.web.cern.ch/ bottom)
Authors: We agree to this change
end of requested changes ==
Best regards,
The authors

Author: Samuel Bein on 2024-12-21 [id 5059]
(in reply to Report 2 on 2024-12-18)Dear Reviewer, Editor,
Thank you for the very helpful review. We have taken all of the comments into account and responded, in some cases with changes to the draft, and in some cases with clarifications/explanations here, all of which is given below.
Reviewer: All the equations and in-line math seem to be written in bold font. Is that done on purpose?
Authors: We also prefer to use non-bold type setting for the equations, if it is fine with all.
Reviewer: Section 1: The introduction contains relatively few references. Alternative or complementary methods to speed up detector simulations should be mentioned here, for instance fast calorimeter simulations with machine learning (see the CaloChallenge paper, 2410.21611, for an overview of different approaches)
Authors: In our view we cover a set of methods whose scope is most similar to our’s, but it is true there is a broader set of ML-based simulation techniques to which we have not referred. To give a broader context, we propose to add two sentence in the introduction: "More generally, ML has been used to replace components of fast simulation, such as the modeling of calorimeter showers, such as the seminal work~\cite{deOliveira:2017pjk}, with a broad review given in~\cite{Krause:2024avx}; similar techniques have been applied in ATLAS fastsim~\cite{ATLAS:2021pzo}. There are also efforts to replace the entire simulation frameworks with generative normalizing flows~\cite{Vaselli:2024hml} in CMS."
Reviewer: Section 2.3: The MDMM method is used to determine the value of the Langrange multiplier between the two loss terms. However, as the authors note, the two loss terms become anti-correlated at some point during the training. The training task is therefore not a constrained optimization, but it has to find a balance between two (potentially) conflicting objectives instead. Please clarify why it is advantageous in this situation to choose the MDMM method instead of a hand-tuned multiplier λ. The latter would also allow for a smoother transition between the two training phases.
Authors: It is true that one could use a hand-tuned multiplier, but the MDMM allows a stable convergence, with damping, to a solution where the unbiased MMD can reach 0 in the first stage. Requiring a constraint on one of the loss terms would not be straightforward because we would need to choose and time-evolve λ ourselves. This method ultimately reduces the number of hyperparameters in the procedure.
Reviewer: Section 4.3: As discussed in the text and shown in Tab. 2, there was almost no effect of the two-stage training compared to the MMD-only training in the LHC example. The authors then argue that optimizing the refinement using distribution matching alone is sufficient in this case because the modes of the fastsim already coincide well with the target distribution. It would be helpful to clarify whether that really means that the bias introduced from the pure distribution matching training is negligible. It could also mean that the first training stage is not effective in improving the correlations between the fast-sim and refined points, because its effects are "forgotten" by the network during the second training stage.
Authors: We can probe to some extent whether information gained during stage 1 is lost during stage 2 in a limited way. The best demonstration that this does not occur is in the evolution of the omniscient MMD throughout both stages of the training (Figure 5), which we see decreases monotonically throughout the training. While there is no evidence that the 2-stage training helps in the Delphes example, it is important to note that this method does not do worse than the MMD-only single stage, and it can always be the case that additional hidden variables exist which are not included in the omniscient MMD, where one would see correlations improving via the 2-stage prescription. However, one may not know if such variables exist in a real world scenario, so we argue it is just better to use the 2-stage training when possible, even though the 1-stage is often enough.
Best regards,
The Authors

---

## Round 1 · Referee Report · Anonymous (Referee 2) · 2024-12-18

Strengths
-
The paper contains a realistic LHC example where the Fast Perfekt refinement leads to significant improvements over the established fast simulation. Improvements are even visible in correlations with hidden features.
-
The neural network architecture is well-motivated as it learns the residuals to the fast-sim features, and evaluating the network is computationally cheap.
-
The analytical example gives a very clear explanation on how the method works.
Weaknesses
-
The introduction does not provide the wider context of other machine learning methods developed to accelerate or improve detector simulations.
-
In the LHC example, it is not clear whether the proposed two-step training procedure really leads to a good balance between accurate modeling of the target distribution and modeling of the correlations between the fast-sim and target distribution.
Report
The method is described clearly and in sufficient detail to be reproduced. There are some aspects that I would like to be clarified in a minor revision (see "Requested changes"), otherwise the acceptance criteria are met.
Requested changes
-
All the equations and in-line math seem to be written in bold font. Is that done on purpose?
-
Section 1: The introduction contains relatively few references. Alternative or complementary methods to speed up detector simulations should be mentioned here, for instance fast calorimeter simulations with machine learning (see the CaloChallenge paper, 2410.21611, for an overview of different approaches)
-
Section 2.3: The MDMM method is used to determine the value of the Langrange multiplier between the two loss terms. However, as the authors note, the two loss terms become anti-correlated at some point during the training. The training task is therefore not a constrained optimization, but it has to find a balance between two (potentially) conflicting objectives instead. Please clarify why it is advantageous in this situation to choose the MDMM method instead of a hand-tuned multiplier $\lambda$. The latter would also allow for a smoother transition between the two training phases.
-
Section 4.3: As discussed in the text and shown in Tab. 2, there was almost no effect of the two-stage training compared to the MMD-only training in the LHC example. The authors then argue that optimizing the refinement using distribution matching alone is sufficient in this case because the modes of the fastsim already coincide well with the target distribution. It would be helpful to clarify whether that really means that the bias introduced from the pure distribution matching training is negligible. It could also mean that the first training stage is not effective in improving the correlations between the fast-sim and refined points, because its effects are "forgotten" by the network during the second training stage.
Recommendation
Ask for minor revision
Dear reviewer, Thank you for the thorough and helpful review, and for considering the paper to make a valuable contribution. We agree with the basis of all of the comments, and propose changes to the paper draft in most cases. We respond and propose below in line. As for the noted point regarding very high-dimensional cases such as the case of refining individual hits within a shower, we have realized the need to clarify the purpose of Fast Perfekt, at least in its presented form, as intended to act on final variables of the simulation and not on intermediate quantities such as hits; we clarify this below in responding to all items. ==
Requested changes
Reviwer: Is there a way to find out which loss to use in a non-toy case? I guess one always wants to include as many variables as possible, so there will not be an omniscient MMD available. I'm asking because I was wondering about the application to calorimeter simulation, where EM showers for the same incident conditions (incident energy / angle) are very similar, but hadronic showers (for example from pions) differ very much from shower to shower. Authors: We propose to tweak the final paragraph to make it more clear that we endorse using the 1-stage, MMD-based training in cases where simulation is refined to match real data, and the 2-stage method is ideal to make fastsim more like fullsim.
For intermediate objects like sim hits from EM showering, the loss terms would likely need to be restructured. Fast Perfekt’s purpose is really to morph the final analysis variables (summary statistics) of objects and events. We propose to add the following sentence in the last paragraph of the conclusions:
“ The refinement acts on final variables or summary statistics of the simulation (i.e., jet properties) rather than intermediate quanities (i.e., calorimeter shower hits).” Reviwer: When comparing corr(x,h) to corr(x^,h′), the results will be skewed if h and h′ differ from each other. I miss a discussion on this in section 2 and also the toy case does not consider h≠h′. Authors: We agree, that we can improve the correlation to the hidden observable, but not to the extent that would be possible if one could also refine h. This case is explored in the Delphes example where the pT and other variables are included in the omniscient MMD but not directly refined.
We propose to add a qualifying statement about correlations to hidden observables: “we also note that the network cannot refine the hidden variable itself, but only correlations to the target hidden variable.” Reviwer: Will the method still work if the Fastsim model suffers from mode-collapse? I'm asking because in the limit of arbitrarily close Fastsim-samples, a deterministic model will not be able to spread the samples apart. Authors: In an extreme case of mode collapse, where all elements of the feature vector default to a given value, Fast Perfket would not be able to refine such an effect away, because the refinement is deterministic and does not add its own noise. However, if some variables (or the ground truth) have not exhibited mode collapse, there is still the possibly to refine. Reviwer: The use of the terms "fastsim" and "fullsim" in the LHC example is misleading, as both are essentially fastsim, just with different parameter cards. Authors: We propose to modify the sentences on 243 for clarity to read: “These events are then processed twice in parallel using Delphes, once with the default CMS detector implementation and treated as the fullsim data set, and once with a ``flawed" implementation yielding the data set we treat as the fastsim.” Minor comments
Reviwer: The abbreviation MDMM is first used below equation 4, but only explained a few paragraphs below that, just above equation 5. Authors: We agree with the minor comments propose to properly define and cite the MDMM when it is first introduced.
Reviwer: The paragraph below equation 6 uses once N, and m
This typo has been corrected Reviwer: The introduction of section 4 refers to "data collected at the LHC", when in fact only simulation is used in the following. Authors: We propose to change “based on” to “tailored to” so it can not be interpreted that we have used real data. Reviwer: GEANT4 asks for 3 references (see https://geant4.web.cern.ch/ bottom) Authors: We agree to this change end of requested changes
== Best regards, Sam, on behalf of the authors

---

## Round 2 · Author Response

We are happy with the review. Most comments have lead to changes to the manuscript, and otherwise we have clarified small point or questions that arose with the Editor/reviewers. We also tweaked the x-axis range in Figure 5 to make the loss evolution more visible in the first few epochs. Many thanks!

---

## Round 2 · List of Changes

Added the sentence in the conclusions: “The refinement acts on final variables or summary statistics of the simulation (i.e., jet properties) rather than intermediate quantities (i.e., calorimeter shower hits).”

Added a qualifying statement: “We also note that the network cannot refine the hidden variable itself, but only correlations to the target hidden variable.”

Modified the sentences to emphasize that the Delphes and modified Delphes are only taken as proxies of fastsim and fullsim, and are not actual fastsim and fullsim: “These events are then processed twice in parallel using Delphes, once with the default CMS detector implementation and treated as the fullsim data set, and once with a ‘flawed’ implementation yielding the data set we treat as the fastsim.”

Moved the definition and citation of MDMM to where it is first introduced (below Equation 4).

Changed “based on” to “tailored to” in the introduction of Section 4 to avoid the impression that real data were used.

Added references for extra ML4Sim context with specific/review-based examples from the literature.

---

## Editorial Decision

published